# Optimizing deep learning models for glaucoma screening with vision transformers for resource efficiency and the pie augmentation method

Sirikorn Sangchocanonta[1], Pakinee Pooprasert[1], Nichapa Lerthirunvibul[1], Kanyarak Patchimnan[1], Phongphan Phienphanich[1], Adirek Munthuli[2], Sujittra Puangarom[1], Rath Itthipanichpong[2,3]*, Kitiya Ratanawongphaibul[2,3], Sunee Chansangpetch[2,3], Anita Manassakorn[2,3], Visanee Tantisevi[2,3], Prin Rojanapongpun[2,3], Charturong Tantibundhit[1]

1 Center of Excellence in Nexus for Advanced Intelligence in Law, Engineering, and Medicine (Nail'Em), Faculty of Engineering, Thammasat School of Engineering, Thammasat University, Pathum Thani, Thailand, 2 Center of Excellence in Glaucoma, Department of Ophthalmology, Chulalongkorn University, Bangkok, Thailand, 3 King Chulalongkorn Memorial Hospital, Thai Red Cross Society, Bangkok, Thailand

* rath.i@chula.ac.th

**Data availability statement:** Data sharing restrictions: There are ethical restrictions on publicly sharing the Chula—Thammasat Glaucoma dataset (GlauCUTU-DATA) used in this study. These restrictions are in place to protect the privacy of human research participants involved in the study. However, we can make a subset of the data available upon request. This subset includes: - Study ID:

## Abstract

Glaucoma is the leading cause of irreversible vision impairment, emphasizing the critical need for early detection. Typically, AI-based glaucoma screening relies on fundus imaging. To tackle the resource and time challenges in glaucoma screening with convolutional neural network (CNN), we chose the Data-efficient image Transformers (DeiT), a vision transformer, known for its reduced computational demands, with preprocessing time decreased by a factor of 10. Our approach utilized the meticulously annotated GlauCUTU-DATA dataset, curated by ophthalmologists through consensus, encompassing both unanimous agreement (3/3) and majority agreement (2/3) data. However, DeiT's performance was initially lower than CNN. Therefore, we introduced the "pie method," an augmentation method aligned with the ISNT rule. Along with employing polar transformation to improved cup region visibility and alignment with the vision transformer's input to elevated performance levels. The classification results demonstrated improvements comparable to CNN. Using the 3/3 data, excluding the superior and nasal regions, especially in glaucoma suspects, sensitivity increased by 40.18% from 47.06% to 88.24%. The average area under the curve (AUC) ± standard deviation (SD) for glaucoma, glaucoma suspects, and no glaucoma were 92.63 ± 4.39%, 92.35 ± 4.39%, and 92.32 ± 1.45%, respectively. With the 2/3 data, excluding the superior and temporal regions, sensitivity for diagnosing glaucoma increased by 11.36% from 47.73% to 59.09%. The average AUC ± SD for glaucoma, glaucoma suspects, and no glaucoma were 68.22 ± 4.45%, 68.23 ± 4.39%, and 73.09 ± 3.05%, respectively. For both datasets, the AUC values for glaucoma, glaucoma suspects, and no glaucoma were 84.53%, 84.54%, and 91.05%, respectively, which approach the performance of a CNN model that achieved 84.70%, 84.69%, and 93.19%, respectively. Moreover, the incorporation of attention maps

A unique identifier for each participant, ensuring anonymity. - Image shape: Details of the image dimensions - Data formats: (1) Fundus images with unanimous agreement by three random well-trained ophthalmologists (3/3). (2) Fundus images with majority agreement (2/3). - Test set: The subset is derived from the test set of the GlauCUTU-DATA, ensuring that the available data are representative of the conditions and findings presented in the study. - Groundtruth and Model Prediction: This subset includes details of the ground truth and the model's predictions to enable independent verification of the accuracy metrics reported in our manuscript. Within this dataset, 'G' represents glaucoma, 'S' represents glaucoma suspect, and 'N' represents no glaucoma. The data used in this study contains sensitive patient information and is owned by a third-party organization, King Chulalongkorn Memorial Hospital, Thai Red Cross Society, Bangkok, Thailand. As a result, the Institutional Review Board (IRB) of the Faculty of Medicine, Chulalongkorn University, Bangkok, Thailand has imposed restrictions on public data sharing. Researchers interested in accessing this dataset can contact the IRB human research office at King Chulalongkorn Memorial Hospital, Thai Red Cross Society using the following information: - Contact: IRB Office, 3rd Floor, Anandamahidol Building - Telephone: +662256 4493/+6698 573 7622 - Email: medchulairb@chula.ac.th - Website: https://irb.md.chula.ac.th.

**Funding:** This work is funded by the Health Systems Research Institute (HSRI) of Thailand (https://www.hsri.or.th), Grant number HSRI. 67-031. The funders have no contributions to the study such as the study design, data collection, and data analysis.

**Competing interests:** The authors have declared that no competing interests exist.

from DeiT facilitated the precise localization of clinically significant areas, such as the disc rim and notching, thereby enhancing the overall effectiveness of glaucoma screening.

# 1 Introduction

Glaucoma, a condition characterized by retinal ganglion cell degeneration and optic nerve head (ONH) damage, ranks as the second most common cause of blindness globally [1,2]. It is estimated that the number of individuals with glaucoma will increase to 111.8 million by 2040 [3]. Visual field loss occurs gradually and is often asymptomatic in the early stages. [2]. Hence, patients remain unaware until later stages when their vision has been affected. Glaucoma causes irreversible blindness so early identification is the most effective treatment [2]. Early diagnosis is particularly beneficial for individuals referred to as glaucoma suspects who have not yet developed symptoms of glaucoma but have risk factors or clinical features including changes in the ONH [4,5]. Consistent monitoring of structural changes in glaucoma suspects is important as progressive structural changes are associated with increased risk of worsening visual losses and often precede visual changes [5]. Identification of these glaucoma suspects will allow for earlier treatment, prevention of visual loss, and better outcomes [2].

Detecting pathological changes in the ONH is essential for early diagnosis, necessitating well-trained ophthalmologists and medical equipment to ensure accurate assessment. Thailand has a total of 1,300 ophthalmologists, resulting in a ratio of 19.1 ophthalmologists per one million people [6]. Most of the ophthalmologists practice in the capital city and larger hospitals, reducing the number of ophthalmologists in rural areas, which are not suitable for prompt detection of glaucoma. Therefore, it is crucial to address the issue of resource limitations to increase healthcare access for early diagnosis of glaucoma.

There are currently various methods available for glaucoma diagnosis and screening including tonometry, visual field (VF) test [7], optical coherence tomography (OCT) [8] and fundus photography [9,10]. Unlike VF test and OCT which are not commonly available in Thailand, the fundus photography is more accessible [11]. As it is widely available, fundus photography is considered an effective method for glaucoma screening by detecting characteristic changes in the ONH [12]. Pathologic features associated with glaucoma include neuroretinal rim thinning, increased optic cup-to-disc ratio, and peripapillary atrophy (PPA) [12]. Most of these pathological features manifest within the ONH region, representing localized information. Therefore, the focus of this study is on utilizing local information from fundus photography.

To further enhance the accuracy and efficiency of glaucoma screening, artificial intelligence (AI) has been introduced to the medical engineering field [9,10,12]. AI shortens the processing time while limiting the potential of human errors. Additionally, various tasks such as segmentation of cup and disc, calculation of the vertical cup-to-disc ratio in the ONH, and extraction of relevant features can be automated with AI [9,13–15]. Early prototypes for glaucoma screening relied on handcrafted features designed by ophthalmologists, a process that was time-consuming and susceptible to human bias [16]. On the contrary, machine learning techniques automatically extract features, making them more scalable and adaptable [17]. Singh et al.'s study exemplifies this with nature-inspired algorithms like the Bat algorithm and particle swarm optimization for retinal image feature selection, achieving accuracy up to 98.95% [18]. Further studies employing the gravitational search optimization algorithm and the integration of algorithms, such as emperor penguin optimization, achieved an accuracy

of above 95% with fewer features [19,20]. These advancements demonstrate the capability of an optimization process in machine learning-based methods in efficient glaucoma classifiaction. However, deep learning techniques, a subset of AI, are more complex and demand larger datasets for effective training, surpassing the capabilities of conventional machine learning algorithms [15,21]. Through leveraging the groundwork of machine learning, deep learning has the potential to enhance efficiency in medical image analysis.

In recent years, deep learning techniques, including Convolutional Neural Networks (CNNs) and vision transformer architectures, have gained attention for their role in glaucoma detection [10,22–24]. CNNs have been widely used in previous studies for their ability to extract spatial features from images. For example, Singh et al. identifies Inception-ResNet-v2 and Xception models as particularly effective across multiple datasets, highlighting the potential in early glaucoma diagnosis and reducing reliance on manual human efforts [25]. Fu et al. introduced Disc-aware Ensemble Network (DENet), a novel approach utilizing ResNet-50 [26] to combine global and local features in fundus images [22]. The DENet consists of four deep streams: global image stream, segmentation-guided network, local disc region stream, and disc polar transformation stream. This integration improves accuracy, sensitivity, and specificity for screening [22]. Phasuk et al. has further developed the DENet by employing DenseNet-121 [27] for classification, simplifying the process by feeding each classification network into a simple artificial neural network (ANN) to obtain the result [10]. Their modification aims to reduce complexity, making it more suitable for low-resource countries [10]. However, both DENet and the method proposed by Phasuk et al. primarily focus on classifying normal and glaucomatous cases, yet their complexity still requires significant computational resources.

In our recent study [28], we extended these efforts by focusing on local features to reduce complexity in the image analysis process. We also considered classifying glaucoma suspects, in addition to normal and glaucomatous cases, while maintaining accuracy. This was achieved through the exploration of newer CNN models like ConvNeXt-S [29]. However, the limitation was the availability of clean and diverse training datasets. Most datasets, especially target populations such as Thai people, have limited or insufficient labels. Therefore, we developed a novel augmentation technique, referred to as the "donut method", using a concept by Hemelings et al. [30]. Unlike traditional techniques such as image rotation, flipping, and randomly cropping, our approach focuses on context-related glaucoma features by combining the ONH and periphery image crops. The augmentation generates images that resemble a donut, which yields promising results for glaucoma screening, especially for glaucoma suspects cases [28].

For a more in-depth analysis of how the model makes decisions and which regions of the image the model focuses on, visual explanations are helpful. CNN architectures require class activation mappings (CAMs) [31], an external resource, to perform visual explanations by accentuating the image parts contributing the most to the activation of specific neurons in the CNNs. Our recent study [28] also included visual explanations using GradCAM++ [32], although such techniques can be influenced by external artifacts and necessitate additional computational resources. Unlike CNNs, vision transformers can create visual explanations on their own using attention maps. The attention maps detect relationships between different image regions which provides insights into the specific areas of focus [33].

An advantage of vision transformers over CNNs is that the model can generate attention map-guided visual explanations on its own which reduces the need for external resources and reduces susceptibility to artifacts, as demonstrated in Fan et al.'s study [23]. The study compared DeiT, a vision transformer model, with ResNet-50 in detecting primary open-angle glaucoma (POAG) [34]. The findings revealed that the vision transformer outperformed

CNNs when applied to external datasets for glaucoma screening [23]. We opted for vision transformers over CNNs as they require fewer resources while delivering performance comparable to that of CNNs.

In this study, polar transformation of a fundus image is used as input for vision transformers by following the ISNT rule [35] when transforming polar coordinates of fundus image back to its Cartesian coordinates. The ISNT rule, used to assess ONH changes, states that in a normal eye, the neuroretinal rim thickness usually follows a specific pattern: the inferior region being the thickest layer, followed by the superior region, nasal region, and then the temporal region being the thinnest layer (inferior > superior > nasal > temporal) [35]. Deviations from this pattern can indicate glaucomatous damage and assist in the diagnosis of the condition [36]. Based on the ISNT rule, we introduced a novel augmentation technique referred to as the "pie method". This method selectively crops out individual pie slices corresponding to each region from the ISNT rule in the polar transformation. By carefully examining the impact of excluding each region on glaucoma screening results, we aim to enhance the model's performance. This study employs the advantages of vision transformer models, including their ability to generate visual explanations, reduce computational demands, and faster processing times, along with "pie method" augmentation technique to enhance performance. Furthermore, the study focuses on identifying glaucoma suspect cases that may not exhibit signs or symptoms of glaucomatous damage but require continuous monitoring of the ONH to prevent irreversible visual defects.

The main contributions of this paper can be summarized as follows:

1. Develop the "pie method", an augmentation technique that enhances the effectiveness of glaucoma screening using the ISNT rule and classifying the shape of each slice as that of a pie.
2. Employ the vision transformer model for glaucoma screening, capitalizing on its benefits, i.e., improved data efficiency, reduced computational resources, and decreased preprocessing time to enhance the model's efficiency and practicality.
3. Utilize the attention map-guided visual explanations generated by the vision transformer model to visualize the areas of interest in fundus images. The attention map replaces the use of CAMs which reduces the occurrences of artifacts from external factors.

The rest of the paper is organized as follows: Sect 2 describes methods, Sects 3 and 4 explain the experimental setups and experimental results, respectively. Followed by Sect 5 with discussion and conclusion. Lastly, future works are mentioned in Sect 6.

## 2 Methods

The study aims to enhance glaucoma screening using fundus images while also learning about the model's decision-making process using attention maps and visual explanations. The proposed architecture employs a DeiT model for classification of glaucomatous, glaucoma suspect, and non-glaucomatous eyes. Furthermore, a novel augmentation method, the "pie method", was introduced to enhance performance. The methods implemented in this study will be explained below, and the summary will be shown in Table 1.

### 2.1 Ensemble classification in our previous study

In our recent study [28], we introduced the donut augmentation method to enhance the performance of glaucoma suspect case classification using local information fundus images

**Table 1. Comparative analysis of advantages and disadvantages of the preceding methods and the proposed method.**

| Method | Advantages | Disadvantages |
|---|---|---|
| Ensemble with 4 models (ConvNeXt-S + Rectangular + Donut) | • Improve accuracy by combining predictions from multiple models<br><br>• The "donut method" improves the performance of glaucoma suspect cases | • Require computationally expensive due to the requirement of training and combining multiple models<br>• Complex to implement and manage |
| Transfer learning with 2 models (ConvNeXt-S + Polar + Donut) | • The "donut method" improves the performance of glaucoma suspect cases<br>• Polar transformation improves the overall performance of classification<br>• Reducing computational resources and time by reducing the need to compile multiple models<br>• Reducing complex to implement | • Require computation resource and time to generate visual explanation with GradCAM++ |
| Transfer learning with 2 models (DeiT-S + Polar + Donut) | • Reducing resource and time for generating visual explanations, as the model can generate attention maps<br>• Reducing computational resources and time by reducing the need to compile multiple models | • The overall classification performance has decreased |
| Transfer learning with 2 models (DeiT-S + Polar + Pie) | • Polar transformation improves the overall performance of classification<br><br>• Reducing resource and time for generating visual explanations, as the model can generate attention maps<br>• Reducing computational resources and time by reducing the need to compile multiple models<br>• The "pie method" improves the overall classification performance | • The overall performance still does not surpass that achieved using ConvNeXt-S |

presented in rectangular form. To achieve this, we employed an ensemble technique that combined the probability outputs of four ConvNeXt-S models. The models were divided into two groups: two models trained with data labeled as unanimous agreement by three random well-trained ophthalmologists (denoted as 3/3) and the other two models using data labeled as majority agreement by two of three random well-trained ophthalmologists (denoted as 2/3). Both groups focused on the classification of 1) glaucoma along with glaucoma suspects (GS) vs. no glaucoma (N), and 2) glaucoma (G) vs. glaucoma suspects (S).

Data labeled as 3/3 are more commonly found in clinical settings and general datasets than the 2/3 data. The rationale behind using both groups of models was to merge the insights gained from models trained on both types of data to create a more robust classification model. This approach increases variability in dataset making it more representative of actual glaucoma screening in a clinical setting. However, this ensemble technique requires simultaneous operation of four models which leads to increased processing times for the classification of glaucoma, glaucoma suspects, and no glaucoma. While this approach showed promising results in enhancing glaucoma suspect classification performance, it can still be improved. Therefore, we pursued further developments to refine our methodology.

## 2.2 Reducing complexity of previous study

Compared to our previous study, this method is more focused on optimizing resource utilization and reducing processing time by minimizing the number of models required and using fundus images in polar form. Ensemble technique is more computationally expensive to train and deploy than individual models because they involve training and combining multiple models. In this study, we presented the local information fundus images in polar form, as opposed to the rectangular form in our previous study. The architecture is kept same with ConvNeXt-S, as well as the classification tasks including the "donut method" [28].

Initially, we trained models with 3/3 data which were then used as pre-trained models for training with 2/3 data and fine-tuning for final classification. This two-step process shortens processing time by using only two models instead of four models in the ensemble. The new method significantly improves the model's performance while reducing computational resource requirements and processing time.

Training the model with fundus images in polar transformation rather than the rectangular form significantly improved performance, as shown in Sect 4. There was an approximately 16.00% boost in sensitivity, from 45.16% to 61.29% for glaucoma suspect, and an approximately 4.00% enhancement in the weighted average F1-score, from 73.19% to 78.08%. However, this approach requires additional resources, particularly for generating GradCAM++ to create visual explanations. As a result, we further developed the ensemble to address this limitation.

## 2.3 Reduce resource to generate visual explanations

In our previous study, we used the ConvNeXt-S, a CNN-based model, which lacks the ability to generate visual explanations on its own. To address this limitation, we employed the DeiT-S, a vision transformer model, wherein its self-attention mechanisms enable them to generate attention maps for visual explanations within the model itself, explained in Sect 2.9.

For further analysis, we evaluated the time complexity of the ConvNeXt-S and DeiT-S models. Critical factors such as the computational resources required for training and their efficiency are reflected in this evaluation. As shown in Fig 1, we illustrated the efficiency of each model in terms of ImageNet-1K accuracy and Giga floating point operations per second (GFLOPs) using the calflops library [37]. ConvNeXt-S achieves a higher accuracy of 83.1%, potentially due to its complex architecture, comprising 50M parameters. However, such complexity demands more computational resources, resulting in higher time complexity and requiring 8.7 GFLOPs for inference. In contrast, DeiT-S prioritizes efficiency with its simpler architecture, consisting of 22M parameters. This results in a lower time complexity of 4.6 GFLOPs and a slightly lower accuracy of 79.8%.

This compromise suggests that DeiT-S, while exhibiting slightly lower accuracy, offers a more efficient utilization of computational resources. Therefore, DeiT-S presents a particular advantage in resource-constrained environments where computational efficiency is of paramount importance. Accordingly, we used the DeiT-S architecture instead of ConvNeXt-S, while maintaining the utilization of fundus images in polar transformation with the "donut method" in augmentation and classification tasks.

## 2.4 Proposed method

In our previous approach, we switched from using ConvNeXt-S to DeiT-S which reduced computational resource demands and processing times. However, we encountered lower

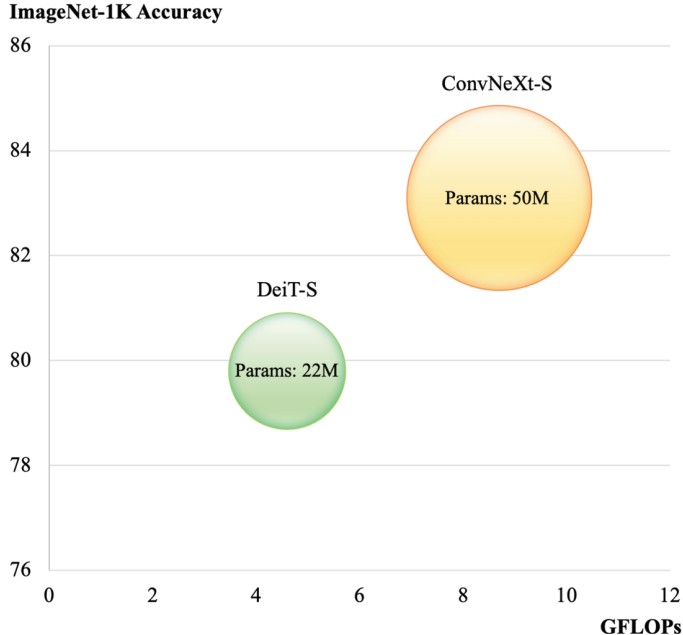

**Fig 1. Comparison of ConvNeXt-S and DeiT-S in terms of correlation between ImageNet-1K accuracy and Giga floating point operations per second (GFLOPs).**

classification performance. To address this challenge, we developed a new method aimed at enhancing performance while retaining the advantages of DeiT-S.

Our solution involved utilizing fundus images in polar transformation. Instead of employing the "donut method" for augmentation, we introduced a novel augmentation approach known as the "pie method". This new method enhances model performance, bringing it closer to the levels achieved by the previous approach. We summarize the comparison of our developed methods in Table 1.

In the subsequent sections (Sects 2.6–2.10), we will provide a detailed explanation of the proposed method and its contributions to our study.

## 2.5 Ethics statement

The dataset utilized in this study was obtained through a retrospective analysis conducted within the medical record system, with approval granted by the Chulalongkorn Institutional Review Board under Application No. 715/61. Given the retrospective nature of the study, individual participants were not required to provide informed consent. However, it is crucial to emphasize that all collected data are fully anonymized. Every effort has been made to safeguard patient privacy and uphold ethical principles.

For research purposes, data were accessed on November 10, 2021, and subsequently stored on the server located at the Center of Excellence in Intelligent Informatics, Speech and Language Technology, and Service Innovation (CILS), Faculty of Engineering, Thammasat School of Engineering, Thammasat University. All the stored data had been de-identified, ensuring that the data cannot be traced back to the individual patient and thus, preserving their privacy. This ensures that information related to patients, such as the Hospital Number (HN), personal history, or any reference numbers, will not be recorded in the database to protect patients' rights in accordance with the Personal Data Protection Act (PDPA) in Thailand. To

maintain anonymity, a random set of five digits has been generated for each image, serving as a unique identifier. The remaining information for each fundus image includes only the eye side (left or right). Details regarding how ophthalmologists in our team diagnose glaucoma will be provided in Sect 3.1.

It is important to note that we cannot access information that could identify individual participants during or after data collection, except for King Chulalongkorn Memorial Hospital, Thai Red Cross Society. Furthermore, this study explicitly excludes the involvement of minors. All data analysis and findings pertain exclusively to adult participants.

## 2.6 Local information

In this study, we utilized fundus images in JPEG format with a resolution of 1,600×1,216 pixels to extract local information from the ONH. To achieve this, we employed our developed segmentation method utilizing the Mask R-CNN model [38]. The performance was evaluated using the Sørensen–Dice coefficients, which yield a score of 96.00% and 92.50% for the optic disc and the optic cup, respectively. The Intersection over Union (IoU) was 89.40% and 81.90% for the optic disc and the optic cup, respectively. These results were considered satisfactory when compared to other studies. In Chakravarty's study, the average Dice coefficient for disc segmentation ranged from 0.87 to 0.97, while for cup segmentation, it was reported as 0.83. Chakravarty's segmentation method outperformed various state-of-the-art techniques across multiple datasets [39]. Additionally, we compared our IoU results to Sevastopolsky's methodology, which revealed an IoU of 0.89 for the disc and a range of 0.69–0.86 for the cup. Sevastopolsky's method demonstrated comparable quality to several state-of-the-art approaches [40].

For padding, we apply a technique based on the centroid point. The left and right padding (Eq 1) and the top and bottom padding (Eq 2) are given by:

$$x_{\text{centroid}} \pm \left( \frac{w_{\max}}{2} + \frac{w_{\max}}{2} \cdot \Delta \right) \tag{1}$$

$$y_{\text{centroid}} \pm \left( \frac{h_{\max}}{2} + \frac{h_{\max}}{2} \cdot \Delta \right), \tag{2}$$

where $w_{\max}$ represents the widest value from the predicted optic disc region, $h_{\max}$ is the highest value from the predicted optic disc region, and $\Delta$ is set to 0.5 for this study.

## 2.7 Polar transformation

To enhance visibility and analysis of relevant ONH features that are primarily radial in nature, we employ a polar transformation on the fundus images [22,41,42]. Utilization of polar transformation is motivated by its ability to effectively represent circular structures and provide a more interpretive representation of the local information. The polar transformation converts the radial relationship between the optic cup and optic disc into a spatial relationship. This transformation results in a structured layer arrangement where the regions corresponding to the cup, disc, and the area outside the ONH are distinctly ordered [41,42]. Fig 2 illustrates the conversion process from the Cartesian coordinate system $(x, y)$ to the polar coordinate system $(r, \theta)$ [43], described by the following equations:

$$x_{\text{d}} = x - x_{\text{c}}, \tag{3}$$

$$y_{\text{d}} = y - y_{\text{c}}, \tag{4}$$

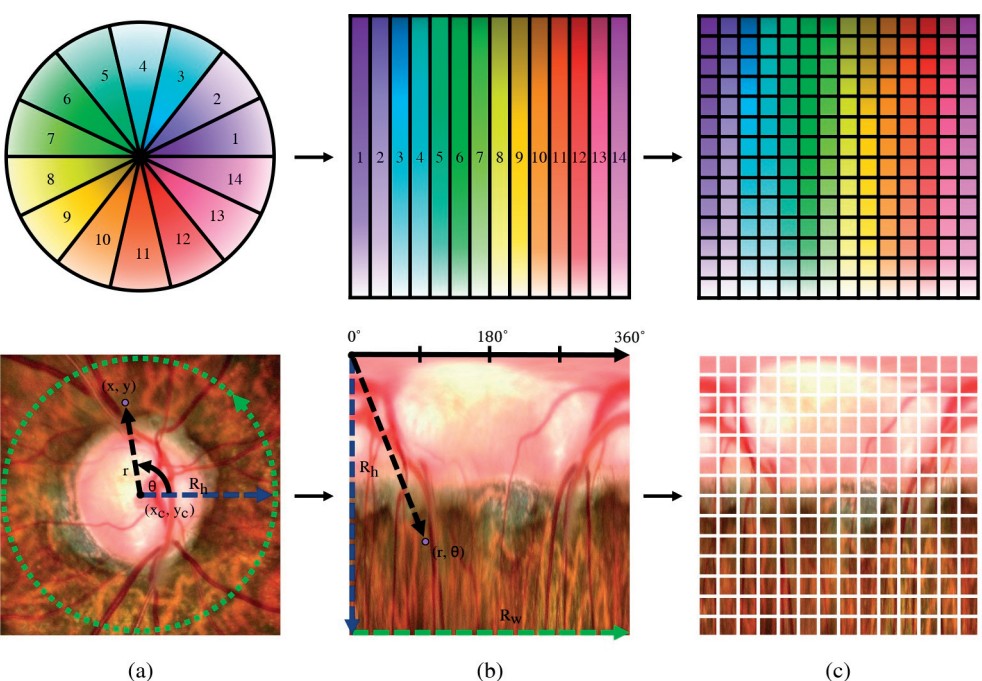

**Fig 2. The polar transformation process of glaucomatous fundus images consists of the following:** a) Cartesian coordinate system, b) transformation of Cartesian coordinates into polar coordinates, and c) polar coordinates as input for vision transformer.

$$r = \sqrt{x_d^2 + y_d^2}, \tag{5}$$

$$\text{and} \quad \theta = \sin^{-1}\left(\frac{y_d}{r}\right), \tag{6}$$

where, $x_d$ represents the horizontal distance between the point $x$ and the centroid point $x_c$, $y_d$ represents the vertical distance between the point $y$ and the centroid point $y_c$, $r$ represents the radial distance from the centroid point, $\theta$ represents the angle, and $R_h$ and $R_w$ represent the height and width of the polar image, respectively.

We used the nasal area as the starting point and ending point for the polar transformation based on the ISNT rule. The ISNT rule is used to assess ONH changes. It states that in a normal eye, the neuroretinal rim thickness follows a specific pattern: the thickest layer is usually in the inferior region, followed by the superior region, nasal region, and finally the thinnest layer in the temporal region [35,36]. To capture the most relevant features associated with glaucoma, we initiated the polar transformation by starting from the nasal part. For right eyes (OD), we moved counterclockwise, while for left eyes (OS), we first applied a horizontal flip and then moved counterclockwise, as illustrated in Fig 2. This approach was designed to ensure comprehensive feature capture in both eye sides.

Applying polar transformation to enhance the performance of classification has demonstrated several advantages compared to using the Cartesian coordinate system [41,42]. Specifically, the polar transformation offers increased visibility of details within the cup region. It results in a more balanced and increased proportion of the cup region compared to the Cartesian coordinate system [22].

Additionally, the polar transformation aligns the fundus image with the input pattern of the vision transformer model. This alignment enables the model to consider each section of the fundus image in accordance with retinal nerve fiber layer (RNFL) thickness sectors and the ISNT rule (inferior, superior, nasal, and temporal quadrants) resembles the pie.

## 2.8 Pie augmentation

One of the objectives was to develop an augmentation method that would enhance the performance of glaucoma classification, specifically in glaucoma suspect cases. To accomplish this, we incorporated augmentations that corresponded with the ISNT rule and the RNFL thickness. This approach aims to provide the model with more relevant and informative features for more accurate predictions.

The concept behind our augmentation method involves selectively cropping out specific regions from the input images to force the model to learn other relevant regions. This approach allows us to extract valuable information from the images that might be overshadowed by dominant features. By selectively cropping out regions, our aim is to highlight various aspects of the image and provide the model with a more comprehensive understanding of the underlying patterns.

As shown in Fig 3, we employed the ISNT rule (inferior, superior, nasal, and temporal) to crop out specific regions in the fundus image for training the model. Deviations from this pattern can indicate glaucomatous damage and assist in the diagnosis of the condition [36]. As the ISNT rule aids in detection of structural glaucomatous damage, we can ensure that the chosen regions for cropping are useful for glaucoma classification. The confusion matrix and the Area Under the Receiver Operating Characteristic Curve (AUC) values were obtained from plotting the Receiver Operating Characteristic (ROC) curve to measure the performance for each cropped-out region. With analysis of the confusion matrix and the AUC values, we were able to identify which crop-out regions contributed to enhancing classification performance [28]. This analysis allowed us to select and augment the data for each class by choosing the regions that significantly improved the model's performance. The cropped-out regions resembled a pie shape, hence the name "pie method".

## 2.9 Vision transformer for glaucoma screening

Vision transformer is based on a self-attention mechanism that allows the network to capture long-range dependencies and relationships between different parts of the fundus image[29]. However, vision transformer's limitation is its dependency on a larger amount of data, and it does not generalize well when trained on insufficient amounts of data. As a result, it becomes computationally intensive and less effective under such circumstances. Recently, DeiT was introduced to address this issue [34]. DeiT employs a distillation process, where a smaller model learns from a larger one in a teacher-student strategy. It utilizes a distillation token to guide knowledge transfer through attention mechanisms, enhancing both model accuracy and training efficiency without the need for extensive data [34].

However, DeiT may not perform as well as CNNs and other vision transformers like ConvNeXt and Swin Transformers [29]. This is because DeiT relies on the self-attention mechanism, which may not be as effective as ConvNeXt and Swin Transformers, which leverage convolutional layers to model spatial relationships between image patches [29,34]. Nevertheless, when we compared DeiT to ConvNeXt and Swin Transformers, we discovered that DeiT's efficient training procedure, driven by the distillation process, allowed it to achieve high accuracy with fewer parameters and less computational demand. Therefore, we chose DeiT model for our study to maximize glaucoma screening accuracy yet minimize the need

Unanimous agreement (3/3)          Majority agreement (2/3)

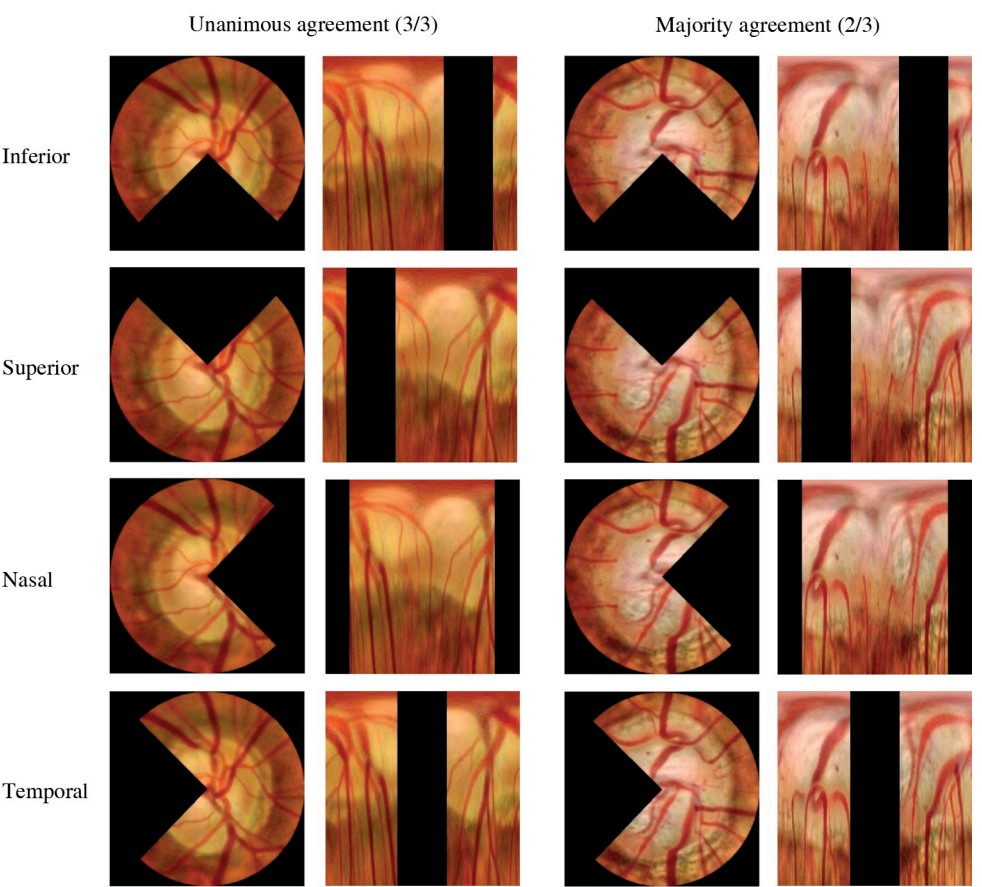

**Fig 3. Examples of glaucomatous fundus images with cropped regions using the "pie method".** The cropped regions correspond to data labeled as unanimous agreement (3/3) and majority agreement (2/3).

for extensive data collection and resource [29]. The implemented DeiT model for classification is illustrated in Fig 4.

We have configured the classification probability output between two cases: 1) glaucoma along with glaucoma suspect (GS) vs. no glaucoma (N) and 2) glaucoma (G) vs. glaucoma suspect (S). To ensure classification probability values from 0 to 1, a Sigmoid activation function [44] was applied in the final layer of the DeiT model. Furthermore, as shown in Fig 4, the model utilizes a default patch size of 16×16. Consequently, the input image with dimensions of 224×224 is divided into a 14×14 grid. Each section of the grid corresponds to a slice of the pie as shown in Fig 2c) and is fed into the model.

### 2.10 Attention map

An attention map is a visual representation of a model's attention weights, highlighting the most informative regions in the input data [33]. These informative regions correspond to the parts of the input that keep the highest attention weights [33]. Attention maps are generated by analyzing the relationships between different image patches or regions, allowing the model to assign different levels of importance to different parts of the input image [33,45].

An advantage of attention maps generated by vision transformers over techniques like Class Activation Maps (CAM) [31] is their reduced susceptibility to artifacts from external

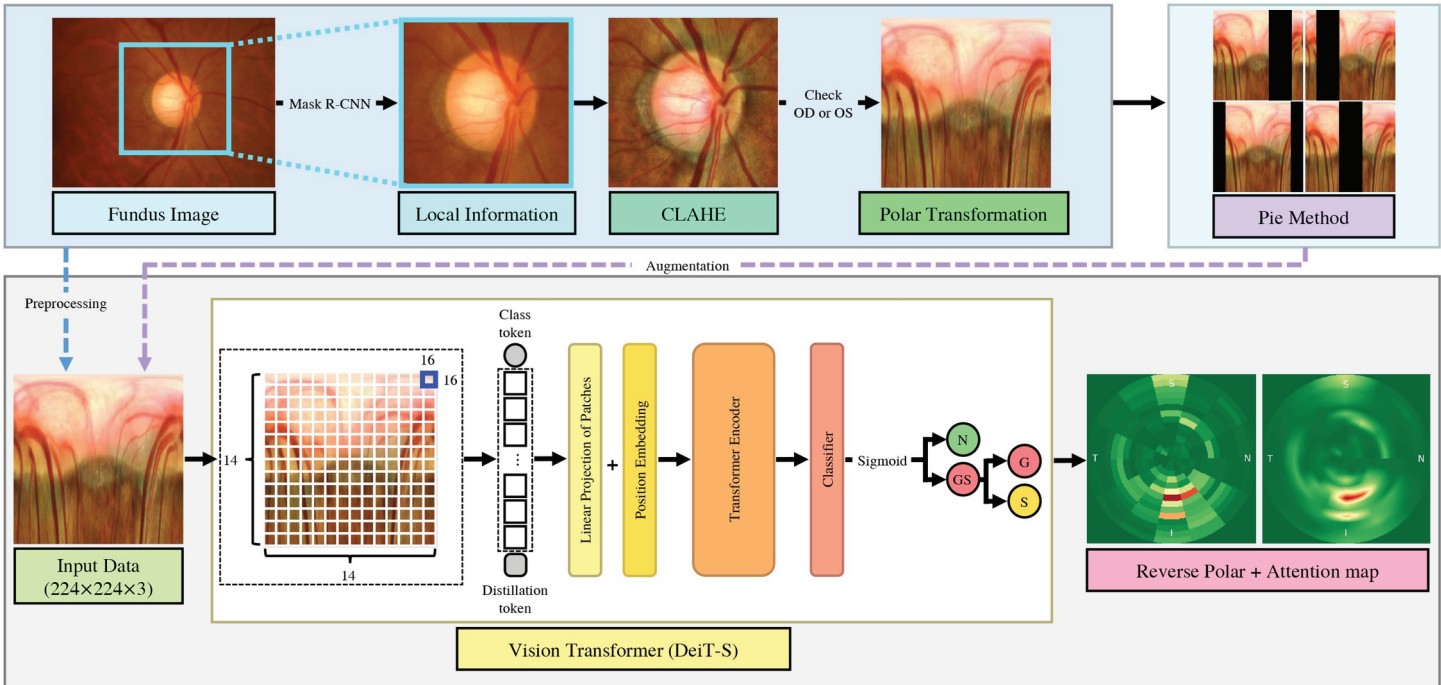

**Fig 4. The process of the proposed glaucoma classification method.**

factors. CAM-based methods can produce artifacts or misleading results due to the reliance on external information. In contrast, attention maps generated by ViT are based on the inherent attention mechanism within the model itself, making them more reliable and less affected by external artifacts [23]. We utilized the attention map generated by the model, along with the gradient values obtained from the classification process, to gain a more comprehensive understanding of the model's decision-making process. We suggest that this information allows localization and identification of the important regions within images.

Self-attention mechanisms in vision transformer models allow the model to prioritize relevant regions by assigning higher attention weights to those patches. By doing so, the model can effectively capture important features and patterns associated with glaucoma, such as optic disc cupping, changes in blood vessel appearance, localized retinal thinning, and PPA [12].

In this study, we employed the DeiT-S model, which consists of 12 attention layers, with each layer containing 6 attention heads. Our approach entailed calculating the gradient as the attention weight for each class and then multiplying it with each head in each layer. Subsequently, we performed head fusion by taking the mean of these values. This process enabled us to generate attention maps for each class, generating visual explanations of the model's predictions.

## 3 Experimental setup

### 3.1 Dataset

The dataset used in this paper was labeled by a group of well-trained ophthalmologists. At least three ophthalmologists (four well-trained ophthalmologists in case of no majority agreement) from the Center of Excellence in Glaucoma, Department of Ophthalmology, Faculty of

**Table 2. GlauCUTU-DATA unanimous agreement (3/3) and majority agreement (2/3) data formats were used, with each data format including the following categories: glaucoma (G), glaucoma suspect (S), and no glaucoma (N).**

| DATA | Unanimous agreement (3/3) | | | | Majority agreement (2/3) | | | |
|---|---|---|---|---|---|---|---|---|
| | G | S | N | Total | G | S | N | Total |
| Training set | 1,302 (23.38%) | 266 (4.78%) | 2,723 (48.89%) | 4,291 (77.04%) | 668 (17.53%) | 680 (17.85%) | 1,558 (40.89%) | 2,906 (76.27%) |
| Validate set | 325 (5.83%) | 67 (1.20%) | 681 (12.23%) | 1,073 (19.26%) | 166 (4.36%) | 171 (4.49%) | 389 (10.21%) | 726 (19.06%) |
| Test set | 86 (1.54%) | 17 (0.31%) | 103 (1.85%) | 206 (3.70%) | 44 (1.15%) | 45 (1.18%) | 89 (2.34%) | 178 (4.67%) |
| Total | 1,713 (30.75%) | 350 (6.28%) | 3,507 (62.96%) | 5,570 (100.00%) | 878 (23.04%) | 896 (23.52%) | 2,036 (53.44%) | 3,810 (100.00%) |

Medicine, Chulalongkorn University, were involved in the labeling process of fundus images. If there is no majority agreement, one of two experienced supervisors would provide the label for that image. This approach ensured accurate labeling for the dataset used in this study.

As shown in Table 2, the dataset consisting of 9,380 fundus images categorized based on a majority vote using two data formats: 1) Fundus images with unanimous agreement by three random well-trained ophthalmologists (3/3) containing 5,570 images (1,713 G eyes, 350 S eyes, and 3,507 N eyes) and 2) Fundus images with majority agreement (2/3) containing 3,810 images (878 G eyes, 896 S eyes, and 2,036 N eyes). For fundus images that did not meet majority agreement, a random supervisor will decide on the final diagnosis. The fundus images dataset is referred to as the Chula–Thammasat Glaucoma dataset (GlauCUTU-DATA), collected in compliance with Chulalongkorn Hospital's Human Research Ethics Certificate No. 715/61.

Furthermore, in order to evaluate the generalizability of our model beyond its initial training scope, we conducted additional testing with a test set from the publicly available DRISHTI–GS1 dataset [46]. This test set includes 51 fundus images, categorized into two groups: 70 glaucomatous and 31 normal optic nerve head images. It is important to note that this data was not employed for model training. Rather, this serves as an independent benchmark for assessing the model's performance on unseen examples. This strategy aids in assessing the model's ability for generalization to unfamiliar data, a critical factor for real-world applications.

## 3.2 Implementation details

In our experimental setup, we employed four DeiT models. These models were developed to classify glaucoma in different scenarios:

1. A model with 3/3 data for classifying between glaucoma along with glaucoma suspect and no glaucoma.
2. A model with 3/3 data for classifying between glaucoma and glaucoma suspect.
3. A model using 2/3 data for classifying between glaucoma along with glaucoma suspect and no glaucoma.
4. A model using 2/3 data for classifying between glaucoma and glaucoma suspects.

The four models were developed in Python using PyTorch backend. The initial parameters were pre-trained on ImageNet [47] for the models trained with 3/3 data. For the models trained with 2/3 data, we used the pre-trained 3/3 data model as a teacher to guide the

training and then fine-tuned to the 2/3 data. To improve the contrast and reduce noise of fundus images, Contrast Limited Adaptive Histogram Equalization (CLAHE) was utilized for pre-processing [48]. Additionally, for the sake of consistency with the input of the pretrained models, each image was resized to a resolution of 224×224×3 (RGB color channels) before fed into the four models.

The objective was to train four models: two models were trained on 3/3 data, while the other two models were trained on 2/3 data. These models were designed for the purpose of classifying the data into three distinct classes. Initially, they grouped glaucoma and glaucoma suspect together, distinguishing them as a single class from no glaucoma. Then, they further classified glaucoma and glaucoma suspect separately. The dataset was divided into training, validation, and testing sets as shown in Table 2. The training data were evenly divided into 5 stratified folds. In each iteration, four of these folds were used for training, and the fifth fold was used as the validation set.

To validate the concept of augmentation with data cropped from different regions following the ISNT rule, the models were trained for 100 epochs with a batch size of 32 in our experiments. The layer-wise adaptive moments based (LAMB) optimizer with $10^{-3}$ learning rate was employed during the training process. Each of the four classification models was trained for a total of 100 epochs with a batch size of 32. The training process utilized the LAMB optimizer with a learning rate set at $10^{-3}$. Our choice of these hyperparameters was the result of thorough experimentation, where we explored a range of values and determined that these settings were indeed optimal. They enabled the training loss and validation loss to converge steadily in the process. The loss function for each model is the Focal Loss, which helps address class imbalance during the training process [49].

## 3.3 Evaluation of performance

The performance of the "pie method" and the classification model was assessed using several evaluation metrics, including AUC, accuracy, sensitivity, specificity, precision, and F1-score.

AUC, a widely used metric in machine learning, measures the overall performance of the model in terms of its ability to correctly classify positive (glaucoma together with glaucoma suspects) and negative (no glaucoma) instances as well as positive (glaucoma) and negative (glaucoma suspects) instances. It quantifies the model's ability to distinguish between different classes. The AUC value was calculated using the following equation [50]:

$$AUC = \int_0^1 ROC(t)\,dt, \tag{7}$$

where ROC(t) is the receiver operating characteristic (ROC) curve at threshold t. The ROC curve is a plot of the True Positive Rate (TPR) against the False Positive Rate (FPR). The TPR is the proportion of positive samples that are correctly classified as positive, and the FPR is the proportion of negative samples that are incorrectly classified as positive.

In addition to AUC, accuracy was computed to assess the overall correctness of the model's predictions [51]. Accuracy was determined using the following equation:

$$Accuracy = \frac{TP + TN}{TP + TN + FP + FN} \times 100\%, \tag{8}$$

where TP represents True Positives, TN represents True Negatives, FP represents False Positives, and FN represents False Negatives, respectively.

Furthermore, sensitivity (also known as recall or TPR) and specificity were calculated to evaluate the model's performance in detecting positive and negative instances, respectively [51]. Sensitivity was calculated using the following equation:

$$\text{Sensitivity} = \frac{\text{TP}}{\text{TP} + \text{FN}} \times 100\%, \tag{9}$$

and specificity was calculated using the following equation:

$$\text{Specificity} = \frac{\text{TN}}{\text{TN} + \text{FP}} \times 100\%, \tag{10}$$

The F1-score was calculated to assess the model's performance in terms of both precision and recall. The F1-score combines these two metrics into a single value that represents the harmonic mean of precision and recall. It is calculated using the following equation:

$$\text{F1} = \frac{2 \times \text{Precision} \times \text{Recall}}{\text{Precision} + \text{Recall}}, \tag{11}$$

where Precision is the ratio of TP to the sum of TP and FP, and Recall is the sensitivity in Eq 9. The Precision is the proportion of correctly predicted positive instances (TP) among all positive predictions.

The F1-score provides a balanced measure of the model's performance, taking into account both precision and recall. It is particularly useful in situations where class imbalance exists. These evaluation metrics provide a comprehensive assessment of the performance of the "pie method" and the classification model, allowing for a thorough analysis of their effectiveness in glaucoma detection and classification.

## 4 Experimental results

### 4.1 Classification results

In this research endeavor, we utilized the "pie method" as a strategic approach to delineate the region of interest for cropping within the augmented training dataset, with our primary objective centered around evaluating its efficacy. This entailed the systematic cropping of individual areas corresponding to the ISNT rule, as illustrated in Fig 3, within the training dataset. Subsequently, distinct models were trained for each of the following areas: 1) Inferior cropped area, 2) Superior cropped area, 3) Nasal cropped area, and 4) Temporal cropped area. Our analysis, as presented in Table 3, unveiled substantial enhancements in the performance of models trained with the 3/3 data when the superior and nasal areas were cropped. Specifically, when we excluded the superior region, we observed a noteworthy increase of 3.56% and 6.13% in the F1-scores for glaucoma and glaucoma suspects, respectively. Although the F1-score for the no glaucoma class exhibited a slight decrement of 0.26%, the micro average F1-score, macro average F1-score, and weighted average F1-score all demonstrated improvements. Consequently, we made the strategic choice to implement the exclusion of the superior region to augment images pertaining to the glaucoma, glaucoma suspects, and no glaucoma categories. Similarly, when we excluded the nasal region, a remarkable 10.50% increase in the F1-score for glaucoma suspects was observed, accompanied by enhancements in the

**Table 3. Performace of "pie method" using GlauCUTU-DATA unanimous agreement (3/3) and majority agreement (2/3).**

| Region | Class | Unanimous agreement (3/3) | | | | | | | | | Majority agreement (2/3) | | | | | | | | |
|---|---|---|---|---|---|---|---|---|---|---|---|---|---|---|---|---|---|---|---|
| | | Acc (%) | Sen (%) | Spec (%) | Prec (%) | F1 (%) | Micro avg. F1 (%) | Macro avg. F1 (%) | Wt. avg. F1 (%) | AUC (%) | Acc (%) | Sen (%) | Spec (%) | Prec (%) | F1 (%) | Micro avg. F1 (%) | Macro avg. F1 (%) | Wt. avg. F1 (%) | AUC (%) |
| Inferior | G | 82.08 (↓6.33) | 67.44 (↓16.28) | 92.06 (↑0.32) | 85.29 (↓2.51) | 75.32 (↓10.39) | | | | 88.55 (↑2.15) | 71.98 (↑1.10) | 50.00 (↑2.27) | 78.99 (↑0.73) | 43.14 (↑1.96) | 46.32 (↑2.11) | | | | 72.28 (↑6.11) |
| | S | 86.32 (↓6.92) | 88.24 (↑41.18) | 86.26 (↓11.22) | 35.71 (↓25.83) | 50.85 (↓2.48) | 78.77 (↓7.22) | 71.76 (↓4.84) | 80.44 (↓5.14) | 87.55 (↑1.24) | 68.13 (↓1.65) | 57.78 (↑22.22) | 71.53 (↓9.49) | 40.00 (↑1.90) | 47.27 (↑10.49) | 52.20 (↓3.29) | 50.90 (↑0.46) | 53.09 (↓2.63) | 72.77 (↑6.58) |
| | N | 89.15 (↓1.19) | 86.24 (↑7.99) | 92.23 (↑5.82) | 92.16 (↑4.66) | 89.10 (↓1.64) | | | | 97.23 (↓3.91) | 64.29 (↓6.04) | 50.54 (↓18.28) | 78.65 (↑6.74) | 71.21 (↓0.70) | 59.12 (↓11.21) | | | | 70.30 (↓1.90) |
| Superior | G | 91.04 (↑2.63) | 91.86 (↑8.14) | 90.48 (↓1.26) | 86.81 (↓0.99) | 89.27 (↓3.56) | | | | 94.01 (↑7.61) | 69.23 (↓1.65) | 65.91 (↑18.18) | 70.29 (↓7.97) | 41.43 (↑0.25) | 50.88 (↑6.67) | | | | 76.63 (↑10.45) |
| | S | 92.92 (↓0.32) | 64.71 (↑17.65) | 95.38 (↓1.99) | 55.00 (↓6.54) | 59.46 (↑6.13) | 87.26 (↑1.27) | 79.73 (↑3.13) | 87.50 (↑1.92) | 94.02 (↑7.01) | 77.47 (↑7.69) | 40.00 (↑4.44) | 89.78 (↑8.76) | 56.25 (↑18.15) | 46.75 (↑9.97) | 58.24 (↑2.75) | 55.28 (↑4.84) | 55.28 (↑2.99) | 76.64 (↑10.45) |
| | N | 90.57 (↑0.23) | 87.16 (↑7.07) | 94.17 (↑7.76) | 94.06 (↑6.56) | 90.48 (↑0.26) | | | | 97.55 (↑4.23) | 69.78 (↓0.55) | 63.44 (↓5.38) | 76.40 (↑4.49) | 73.75 (↑1.84) | 68.21 (↓2.12) | | | | 74.96 (↑2.75) |
| Nasal | G | 88.68 (↑0.27) | 81.40 (↓2.32) | 93.65 (↑1.91) | 89.74 (↑1.94) | 85.37 (↓0.34) | | | | 95.13 (↑8.73) | 75.27 (↑4.39) | 47.73 (0.00) | 84.06 (↑5.80) | 48.83 (↑7.65) | 48.28 (↑4.07) | | | | 73.07 (↑6.89) |
| | S | 91.98 (↓11.26) | 88.24 (↑41.18) | 92.31 (↓5.06) | 50.00 (↓11.54) | 63.83 (↑10.50) | 84.91 (↓1.08) | 79.47 (↑2.87) | 85.61 (↑0.03) | 95.11 (↑8.79) | 67.03 (↓2.75) | 44.44 (↑8.88) | 74.45 (↓6.57) | 36.36 (↓1.74) | 40.00 (↓3.22) | 54.40 (↓1.09) | 51.27 (↑0.83) | 55.05 (↓6.57) | 73.07 (↑6.88) |
| | N | 89.15 (↓1.19) | 87.16 (↑7.07) | 91.26 (↓4.85) | 91.35 (↓3.85) | 89.20 (↓1.54) | | | | 97.72 (↑4.40) | 66.48 (↓3.85) | 62.37 (↓6.45) | 70.79 (↓1.12) | 69.05 (↑2.86) | 65.54 (↓4.79) | | | | 72.91 (↓0.70) |
| Temporal | G | 87.74 (↓0.67) | 88.37 (↑4.65) | 87.30 (↓4.44) | 82.61 (↓5.19) | 85.39 (↓0.32) | | | | 92.52 (↑6.12) | 71.43 (↑0.55) | 47.73 (0.00) | 78.99 (↑0.73) | 42.00 (↑0.82) | 44.68 (↑0.47) | | | | 66.44 (↑0.27) |
| | S | 92.45 (↓0.79) | 52.94 (↑5.88) | 95.90 (↓1.47) | 52.94 (↓8.60) | 52.94 (↓0.39) | 84.43 (↓1.56) | 75.67 (↓0.93) | 84.48 (↓1.10) | 92.51 (↓6.20) | 75.27 (↑5.49) | 40.00 (↑4.44) | 86.86 (↑5.84) | 50.00 (↑11.90) | 44.44 (↑7.66) | 57.69 (↑2.20) | 52.99 (↑2.55) | 57.48 (↑1.76) | 66.43 (↑0.25) |
| | N | 88.68 (↓1.66) | 86.24 (↓7.99) | 91.26 (↑4.85) | 91.26 (↓3.76) | 88.68 (↓2.06) | | | | 97.23 (↓3.91) | 68.68 (↓1.65) | 70.97 (↓2.15) | 66.29 (↓5.62) | 68.75 (↓3.16) | 69.84 (↓0.49) | | | | 76.67 (↑4.46) |

macro average F1-score and weighted average F1-score. Nevertheless, it is crucial to acknowledge that the F1-scores for glaucoma and no glaucoma, as well as the micro average F1-score, experienced a decline. Consequently, we concluded that the exclusion of the nasal region would be advantageous in augmenting images specifically for the glaucoma suspect category.

In the case of the model trained with 2/3 data, similar to the direction taken with the 3/3 model, we observed significant improvements when excluded the superior and temporal regions. Specifically, the F1-scores for glaucoma and glaucoma suspects increased by 6.67% and 9.97%, respectively, when we excluded the superior region. Similarly, cropping out the temporal region led to an increase of 0.47% and 7.66% in the F1-scores for glaucoma and glaucoma suspects, respectively. It is worth noting that the F1-score for no glaucoma saw a slight decrease of 2.12% and 0.49% for the superior and temporal regions, respectively. However, the micro average F1-score, macro average F1-score, and weighted average F1-score all exhibited increases. Consequently, we decided to apply the cropping out of both regions to augment the images for the glaucoma, glaucoma suspects, and no glaucoma classes. Regarding the inferior and nasal regions, although the F1-scores for glaucoma and glaucoma suspects, as well as the macro average F1-scores, increased, we noticed a decrease in the weight average F1-score. This decrease was primarily due to data imbalance. Considering the importance of the weighted average F1-score, we decided not to apply augmentation using these regions.

Following our experimentation with the "pie method", we carefully considered which regions to crop out for augmentation. Once we made these selections, we proceeded to augment the data. Our training approach involved initially training the model with 3/3 data and then continuing the training process with 2/3 data.

To evaluate the effectiveness of our proposed method, we compared it with the method we developed earlier, as mentioned in Sect 2. The results are presented in Table 4. Notably, for 3/3 data, our proposed method demonstrated a significant improvement in overall performance. It was particularly effective in enhancing the performance of glaucoma suspect classification, increasing sensitivity by approximately 20%, from 64.71% to 88.24%, and improving the F1-score from 56.41% to 78.94% when using the "pie method" instead of the "donut method",

**Table 4. Results of the proposed "pie method" were compared to different methods on 3/3 data; accuracy, sensitivity, specificity, precision, F1-score, micro average F1-score, macro average F1-score and weight average F1-score were evaluated with a test set, and average AUC ± SD was evaluated with a validated set.**

| Model | Class | Unanimous agreement (3/3) | | | | | | | | |
|---|---|---|---|---|---|---|---|---|---|---|
| | | Acc (%) | Sen (%) | Spec (%) | Prec (%) | F1 (%) | Micro avg. F1 (%) | Macro avg. F1 (%) | Wt. avg. F1 (%) | Avg. AUC ± SD (%) |
| ConvNeXt–S + Local + Donut | G | 91.75 | 90.70 | 92.50 | 89.66 | 90.17 | | | | 90.95 ± 3.40 |
| | S | 93.69 | 70.59 | 95.77 | 60.00 | 64.86 | 90.78 | 83.69 | 91.02 | 91.35 ± 3.82 |
| | N | 96.12 | 94.17 | 98.06 | 97.98 | 96.04 | | | | 98.06 ± 0.38 |
| ConvNeXt–S + Polar + Donut | G | 95.63 | 94.19 | 96.67 | 95.29 | 94.74 | | | | 90.20 ± 3.44 |
| | S | 96.12 | 94.12 | 96.30 | 69.57 | 80.00 | 93.20 | 89.75 | 93.42 | 90.34 ± 3.28 |
| | N | 94.66 | 92.23 | 97.09 | 96.94 | 94.53 | | | | 97.86 ± 0.33 |
| DeiT–S + Polar + Donut | G | 91.26 | 87.21 | 94.17 | 91.46 | 89.29 | | | | 89.10 ± 2.73 |
| | S | 91.75 | 64.71 | 94.18 | 50.00 | 56.41 | 86.41 | 78.48 | 86.81 | 88.94 ± 3.92 |
| | N | 89.81 | 89.32 | 90.29 | 90.20 | 89.76 | | | | 90.96 ± 1.83 |
| DeiT–S + Polar + Pie | G | 94.71 | 93.02 | 95.00 | 93.02 | 93.02 | | | | 92.63 ± 4.39 |
| | S | 96.12 | 88.24 | 96.83 | 71.43 | 78.95 | 92.72 | 89.01 | 92.87 | 92.35 ± 4.39 |
| | N | 95.15 | 93.20 | 97.09 | 96.97 | 95.05 | | | | 92.32 ± 1.45 |

with the same DeiT-S architecture. Furthermore, when we compared our proposed method with the ConvNeXt-S architecture, it outperformed our recent study [28] and achieved performance levels close to the method using polar form, which achieved weighted average F1-score of 91.02% and 93.42%, respectively. Our new proposed method achieved an impressive 92.87%.

For the 2/3 data as shown in Table 5, our proposed method did not exhibit a significant increase in performance when compared to our previous method. The results were quite similar to those obtained by other methods, with a weighted average F1-score of 63.30% for our proposed method. This result was comparable to our recent study [28] and the method using polar form with the ConvNeXt-S architecture, which achieved weighted average F1-score of 59.67% and 60.45%, respectively.

When evaluating our proposed method for its applicability in actual screening, we considered all data, including both 3/3 and 2/3 data combined, as shown in Table 6. The results demonstrated that our proposed method exhibited an approximately 6% increase in the weighted average F1-score, rising from 68.54% to 74.74%, when compared to the "pie method" instead of the "donut method", while using the same DeiT architecture. However, when comparing our proposed method to ConvNeXt-S, it did not surpass the performance but came quite close. Our recent study [28] and the method using polar form with the ConvNeXt-S architecture, we achieved weighted average F1-scores of 73.19% and 78.08%, respectively. Furthermore, when focusing on glaucoma and glaucoma suspect classification, our proposed method outperformed our recent study [28]. It improved the F1-score from 61.54% to 87.69% for glaucoma and from 45.16% to 51.61% for glaucoma suspects.

In addition, as shown in Table 7, our evaluation ensured the proposed method was not trained on the DRISHTI–GS1 test set. This approach allows us to determine how well the model performs on unseen data. Our results are promising, achieving an F1-score of 93.33%, accuracy of 90.20%, and sensitivity of 93.11%. While there is a chance at improvement, these results indicate the generalizability and potential of our method for real-world applications in glaucoma classification.

**Table 5. Results of the proposed "pie method" were compared to different methods on 2/3 data; accuracy, sensitivity, specificity, precision, F1-score, micro average F1-score, macro average F1-score and weight average F1-score were evaluated with a test set, and average AUC $\pm$ SD was evaluated with a validated set.**

| Model | Class | Majority agreement (2/3) | | | | | | | | |
|---|---|---|---|---|---|---|---|---|---|---|
| | | Acc (%) | Sen (%) | Spec (%) | Prec (%) | F1 (%) | Micro avg. F1 (%) | Macro avg. F1 (%) | Wt. avg. F1 (%) | Avg. AUC $\pm$ SD (%) |
| ConvNeXt–S + Local + Donut | G | 74.72 | 61.36 | 79.10 | 49.09 | 54.55 | | | | 75.39 $\pm$ 2.13 |
| | S | 73.60 | 42.22 | 84.21 | 47.50 | 44.71 | 59.55 | 56.34 | 59.67 | 75.35 $\pm$ 2.13 |
| | N | 70.79 | 67.42 | 74.16 | 72.29 | 69.77 | | | | 78.41 $\pm$ 1.69 |
| ConvNeXt–S + Polar + Donut | G | 71.35 | 59.09 | 75.37 | 44.07 | 50.49 | | | | 74.25 $\pm$ 1.53 |
| | S | 75.28 | 53.33 | 82.71 | 51.06 | 52.17 | 59.55 | 57.41 | 60.45 | 74.22 $\pm$ 1.54 |
| | N | 72.47 | 62.92 | 82.02 | 77.78 | 69.57 | | | | 77.84 $\pm$ 3.03 |
| DeiT–S + Polar + Donut | G | 70.79 | 72.73 | 70.15 | 44.44 | 55.17 | | | | 67.46 $\pm$ 5.62 |
| | S | 76.40 | 28.89 | 92.48 | 56.52 | 38.24 | 61.24 | 55.94 | 60.51 | 67.37 $\pm$ 4.60 |
| | N | 75.28 | 71.91 | 78.65 | 77.11 | 74.42 | | | | 72.11 $\pm$ 1.17 |
| DeiT–S + Polar + Pie | G | 77.53 | 59.09 | 83.58 | 54.17 | 56.52 | | | | 68.22 $\pm$ 4.45 |
| | S | 74.72 | 44.44 | 84.96 | 50.00 | 47.06 | 63.48 | 59.48 | 63.30 | 68.23 $\pm$ 4.39 |
| | N | 74.72 | 75.28 | 74.16 | 74.44 | 74.86 | | | | 73.09 $\pm$ 3.05 |

**Table 6. Results of the proposed "pie method" were compared to different methods on 3/3 and 2/3 data; accuracy, sensitivity, specificity, precision, F1-score, micro average F1-score, macro average F1-score and weight average F1-score were evaluated with a test set, and AUC was evaluated with a validated set.**

| Model | Class | All data (3/3) + (2/3) | | | | | | | | |
|---|---|---|---|---|---|---|---|---|---|---|
| | | Acc (%) | Sen (%) | Spec (%) | Prec (%) | F1 (%) | Micro avg. F1 (%) | Macro avg. F1 (%) | Wt. avg. F1 (%) | AUC (%) |
| ConvNeXt–S + Local + Donut | G | 80.73 | 61.54 | 90.55 | 76.92 | 68.38 | | | | 87.55 |
| | S | 85.16 | 45.16 | 92.86 | 54.90 | 49.56 | 74.22 | 67.34 | 73.19 | 87.55 |
| | N | 82.55 | 92.19 | 72.92 | 77.29 | 84.09 | | | | 91.84 |
| ConvNeXt–S + Polar + Donut | G | 83.33 | 81.54 | 84.25 | 72.60 | 76.81 | | | | 84.70 |
| | S | 85.42 | 61.29 | 90.06 | 54.29 | 57.58 | 77.60 | 73.31 | 78.08 | 84.69 |
| | N | 86.46 | 80.21 | 92.71 | 91.67 | 85.56 | | | | 93.19 |
| DeiT–S + Polar + Donut | G | 80.47 | 55.38 | 93.31 | 80.90 | 65.75 | | | | 78.57 |
| | S | 71.09 | 51.61 | 74.84 | 28.32 | 36.57 | 66.41 | 61.02 | 68.54 | 78.54 |
| | N | 81.25 | 78.65 | 83.85 | 82.97 | 80.75 | | | | 92.35 |
| DeiT–S + Polar + Pie | G | 82.29 | 87.69 | 79.53 | 68.67 | 77.03 | | | | 84.53 |
| | S | 83.85 | 51.61 | 90.06 | 50.00 | 50.79 | 74.48 | 69.58 | 74.74 | 84.54 |
| | N | 82.81 | 72.92 | 92.71 | 90.91 | 80.92 | | | | 91.05 |

**Table 7. Results of testing our proposed method on DRISHTI-GS1 were compared with previous studies that classify glaucoma in fundus images.**

| Model | All data (3/3) + (2/3) | | | | | | | | |
|---|---|---|---|---|---|---|---|---|---|
| | Acc (%) | Sen (%) | Spec (%) | Prec (%) | F1 (%) | Micro avg. F1 (%) | Macro avg. F1 (%) | Wt. avg. F1 (%) | AUC (%) |
| Nirmala et al. [52] | 94.4 | 91 | - | - | - | - | - | - | - |
| Singh et al. [25] | | | | | | | | | |
| Inception-ResNet-v2 | 91.08 | - | - | - | - | - | - | - | 90.87 |
| Xception | 85.14 | - | - | - | - | - | - | - | 83.89 |
| DenseNet-201 | 84.15 | - | - | - | - | - | - | - | 82.28 |
| Inception-ResNet-v2 (FT + TL) | 91.08 | - | - | - | - | - | - | - | 90.87 |
| Fang et al. [53] | 98.90 | 99.17 | - | 99.31 | 99.24 | - | - | - | 94 |
| Our proposed method | 90.20 | 92.11 | 84.62 | 94.59 | 93.33 | 90.20 | 87.41 | 90.31 | 86.54 |

## 4.2 Visual explanation results

To gain deeper insights into the classification process, attention maps were utilized. Attention maps allowed us to visualize the specific areas within the input images that the model focused on when making its classification decisions. Additionally, by analyzing the attention maps alongside the gradient values obtained from the classification results, we were able to identify the regions of interest that played a pivotal role in the classification process. Figs 5 and 6 present a set of fundus images of glaucoma, glaucoma suspects, and no glaucoma cases. These images have been transformed from polar to Cartesian coordinates for easier interpretation and comparison comparison between the GradCAM++ method employed by ConvNeXt-S with the "donut method" and the attention maps generated by DeiT-S using the "pie method".

In our discussions with the ophthalmologists on our team, based on the results from Fig 5(a), which represents TP of glaucoma from 3/3 data, revealed that GradCAM++ focus extended beyond the laminar dot to notching (A) and showed excessive interest to the PPA (A and B). In contrast, the attention map showed an advantage by focusing more on clinical points of interest, particularly in both C and D regions in the superior and inferior areas. For C, the attention map highlighted the thinning of neuroretinal rim. Similarly, region D also exhibited attention to the notching.

In Fig 5(b), representing TP of glaucoma from 2/3 data, GradCAM++ demonstrated a focus on the superior thinning of the neuroretinal rim, including the bayonet sign at 12 o'clock in area A. The bayonet sign in this region was distinctly evident in the image. In contrast, the attention map showed a focus on the inferior thinning rim in area B. Additionally, the attention map pinpointed a small area of the bayonet sign in area C, which was notably smaller and adjacent to area A, as observed in GradCAM++.

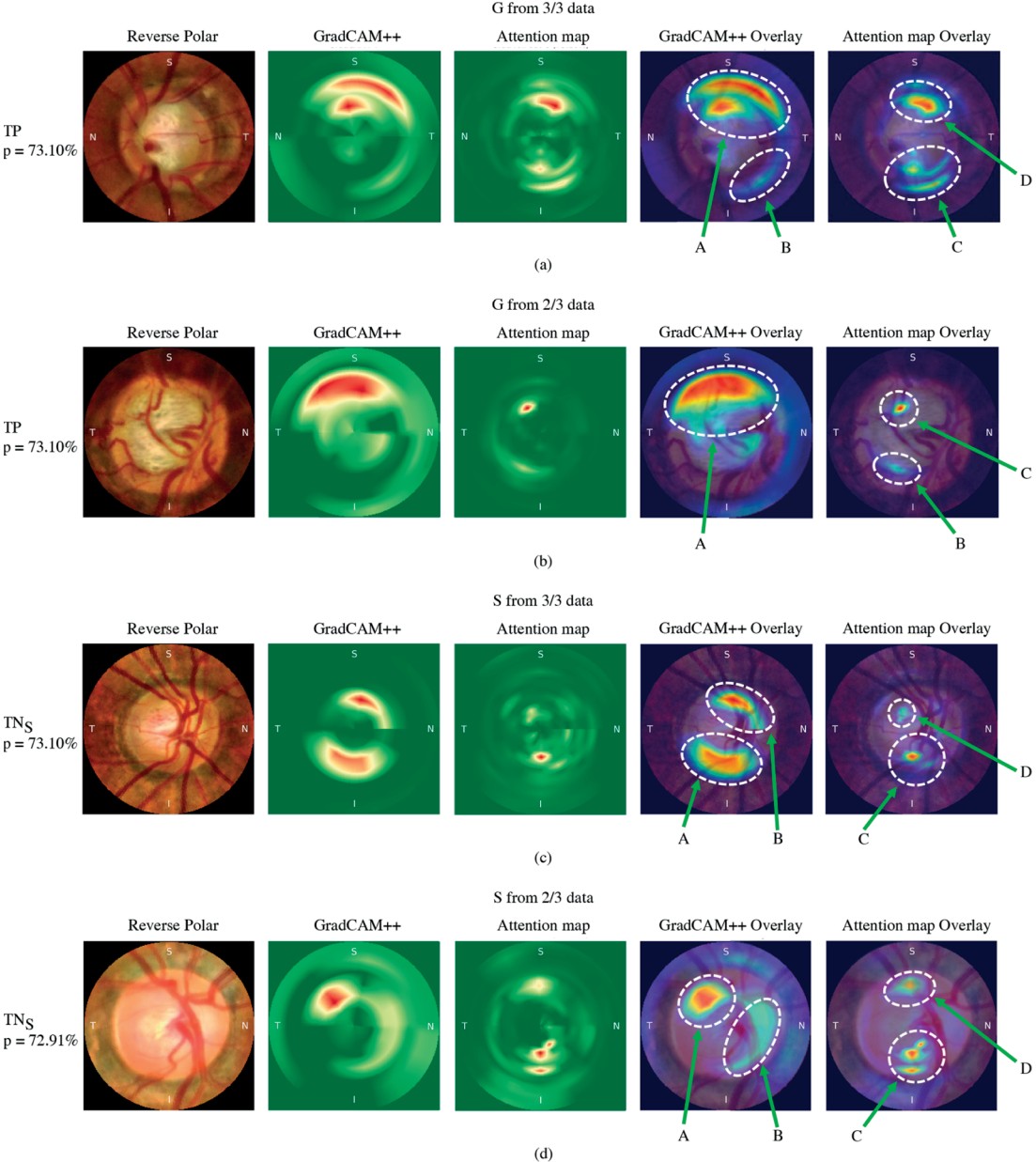

**Fig 5. Fundus images with glaucoma and glaucoma suspect through CLAHE with reverse polar transformation, Grad-CAM++ heatmap of ConvNeXt-S model trained on polar with "donut method", attention map of DeiT- S model trained on polar with "pie method" with gradient of predicted class, GradCAM++ heatmap overlay on reverse polar and attention map overlay on reverse polar.**

Moving on to Fig 5(c), represents TN of glaucoma suspect from 3/3 data. GradCAM++ highlighted the thinning part in A and B, with its focus covering a larger area. In contrast, the attention map in C focused on an irregularly curved vessel in a more localized area than GradCAM++ in A, and in D, it highlighted the laminar dot.

Fig 5(d), representing TN of glaucoma suspect from 2/3 data, GradCAM++ exhibited interest in areas A and B, instead of the areas with notching and the intersection of blood vessels in C and D that the attention map focused on. This figure shows that the attention map outperformed GradCAM++.

Fig 6(a), which represents TN of no glaucoma without other pathologies from 3/3 data, GradCAM++ highlighted the neuroretinal rim, indicating its normal thickness (A). The attention map focused on the same rim but exhibited a more localized effect in B. However, GradCAM++ highlighted a larger area that did not show any abnormal signs.

Fig 6(b), represents TN of no glaucoma without other pathologies from 2/3 data, followed a similar pattern as (a), focusing on the same area, i.e., neuroretinal rim (A and B). However, the attention map exhibited a more localized effect in this case.

In Fig 6(c) and 6(d), representing TN of no glaucoma with other pathologies cases from 2/3 data, these figures showed consistent patterns. GradCAM++ exhibit the spread in the vessel area for both A and B in (c) and A in (d) when compared to the attention map. Furthermore, the attention map provides a clearer and more localized representation for both C and D in (c) and B in (d).

The results of this comparison between GradCAM++ and the attention maps revealed that the attention maps with gradients highlighted more localized regions without scattering. By incorporating attention maps and gradient values, our approach offers enhanced interpretability and localization of the model's decision-making process.

## 5 Discussion and conclusion

In an effort to streamline time and resource demands, we investigated the feasibility of employing the DeiT-S model. While initial attempts using DeiT-S with polar transformation and the "donut method" yielded unsatisfactory results, we persevered by applying the "pie method" and inputting the data into the model. The outcomes, as illustrated in Table 6, reveal that transitioning from ConvNeXt-S to DeiT-S while retaining the same "donut method" led to diminished performance. Our objective is to maintain the benefits of reduced time and resource consumption while enhancing performance to levels comparable to the previously employed ConvNeXt-S model. To achieve this goal, we further refined our proposed approach, which involves leveraging the DeiT-S model with the "pie method" and the polar transformation of fundus images as input. This refined methodology achieved performance metrics comparable to the earlier established techniques. Although our proposed method did not exceed the performance of methods employing the ConvNeXt-S model, which attained a weighted average F1-score of 78.08%, it came close, reaching a weighted average F1-score of 74.74%.

Additionally, we evaluated the real-world performance of the ConvNeXt-S and DeiT-S models. By measuring the actual time taken to complete the practical test of classifying a single fundus image and producing a heat map, we gain insights into their efficiency and suitability for deployment. Table 8 displays the results of the performance test conducted on both central processing unit (CPU) and graphics processing unit (GPU). The tests on GPU portray that ConvNeXt-S takes approximately 2 seconds, while DeiT-S takes merely 0.2 seconds, offering a tenfold reduction in the actual time required for classifying results and generating heat

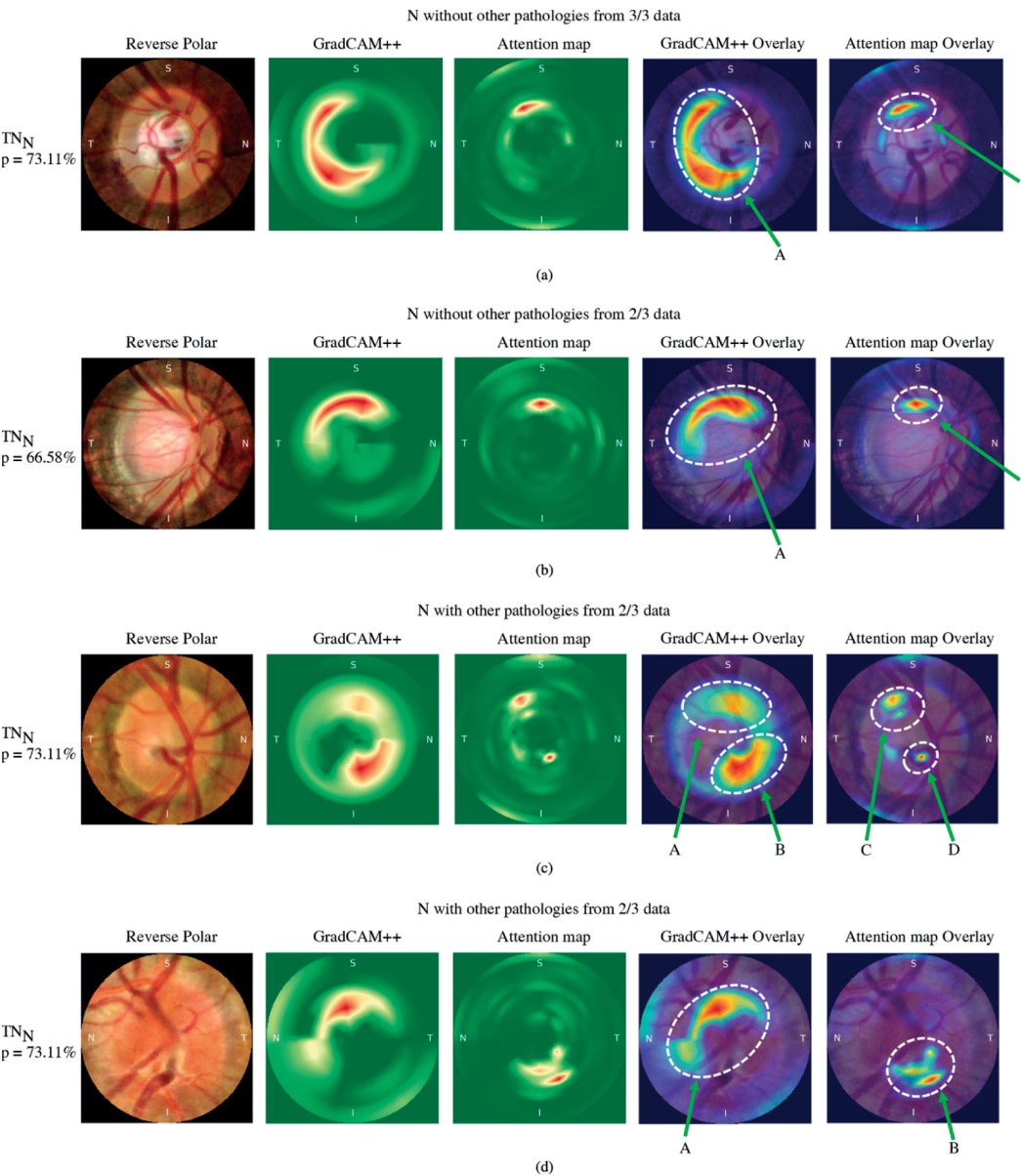

**Fig 6. Fundus images with no glaucoma through CLAHE with reverse polar transformation, GradCAM++ heatmap of ConvNeXt-S model trained on polar with "donut method", attention map of DeiT-S model trained on polar with "pie method" with gradient of predicted class, GradCAM++ heatmap overlay on reverse polar and attention map overlay on reverse polar.**

maps. On the CPU, both ConvNeXt-S' and DeiT-S' ability to complete tasks in less than a second, with DeiT-S being faster. The tests on CPU are especially applicable for virtual machines in hospitals across various regions where resource consideration is critical. Hence, our proposed approach offers a significant reduction in computational time, enhancing the models' capability to classify results for individual fundus images expeditiously.

Our novel augmentation technique, termed the "pie method," has demonstrated enhanced performance in the context of glaucoma screening, particularly in cases involving glaucoma suspects. Through the selective removal of regions of interest within images, these methods

**Table 8. Comparison of actual time taken by ConvNeXt–S and DeiT–S on central processing unit (CPU) and graphics processing unit (GPU).**

| Model | Time on CPU (s) | Time on GPU (s) |
|---|---|---|
| ConvNeXt–S + GradCAM++ | 2 | 0.3 |
| DeiT–S + Attention map | 0.2 | 0.003 |

have proven effective in refining the precision of the screening process. This is particularly significant in situations where early detection and precise diagnosis hold paramount importance for the efficacy of subsequent treatment. In Table 3, we observe that the exclusion of the superior region yields a substantial improvement in the performance across glaucoma, glaucoma suspect, and no glaucoma cases. This improvement is evident in the weighted average F1-score, which registers an increase of 1.92% for 3/3 data and 2.99% for 2/3 data. Furthermore, both micro and macro average F1-scores experience favorable outcomes with this approach. By excluding the superior region, the model's attention is deliberately directed towards the inferior and temporal areas, which are the predominant locations for glaucomatous optic nerve damage resulting from elevated intraocular pressure.

Conversely, the exclusion of the nasal region results in a notable increase of 10.50% in the F1-score for 3/3 data, primarily bolstering the weighted average F1-score. This enhancement can be attributed to the fact that the nasal region typically exhibits fewer discernible signs of glaucoma. Eliminating this region encourages the model to concentrate its attention on other areas housing more prominent glaucoma indicators, thus enhancing its capability to discriminate between glaucoma and glaucoma suspect cases. However, when dealing with 2/3 data, which tend to possess greater ambiguity and include other pathologies compared to 3/3 data without other pathologies, the exclusion of the nasal region might diminish the available information for the model. This reduction in information could potentially undermine the model's proficiency in distinguishing among glaucoma, glaucoma suspect, and no glaucoma cases, as crucial cues within the nasal region may hold significance in resolving such ambiguous cases. Lastly, the exclusion of the temporal region for 2/3 data leads to a 1.76% improvement in the weighted average F1-score. This enhancement, coupled with other performance metrics, may be attributed to the model being compelled to glean knowledge from alternative regions, considering that a substantial portion of information in the temporal region could be perceived as extraneous noise. By refocusing its attention on other regions, the model gains access to more pertinent information, a particularly valuable asset when handling data fraught with ambiguity.

One notable advantage of using "pie method" is that the resulting visualizations from the models tend to focus more localize on clinically relevant features. This is evident when comparing the attention map with GradCAM++ of previous method in Figs 5 and 6. Although some images in Fig 5(b), 5(c), and Fig 6(a) demonstrate GradCAM++ showing a broader interest in clinical points, it does so by focusing on larger areas and covering nearby regions that it finds interesting. However, this broad focus can lead to a loss of concentration and a spread of attention over the entire area. In contrast, the attention map focuses on smaller, more localized areas, maintaining a neutralized attentional focus. It appears that the attention map is selective, focusing only on areas it deems critically important while completely ignoring areas it considers unimportant, making more efficient use of resources.

Further analysis involves investigating FN in glaucoma cases and FP in glaucoma suspect and no glaucoma cases. In the case of FN in glaucoma, we observed that most were misclassified as glaucoma suspects. The attention maps primarily focused on vessels in the optic disc

and cup region, rather than other areas. This pattern continued in cases where glaucoma was wrongly predicted as no glaucoma, with a predominant focus on vessels.

In the context of FP cases for glaucoma suspects, the pattern remains consistent, as they are mostly misclassified as glaucoma. These cases often emphasize thinning of the neuroretinal rim in the inferior region, which is a characteristic sign of glaucoma. Occasionally, they also exhibit an interest in the PPA in the inferior and temporal regions, as opposed to the thinning rim. Conversely, when FP occurred among glaucoma suspects, misclassifying them as no glaucoma, it was because the model showed a keen interest in the neuroretinal rim in the inferior area, which, in these cases, did not display signs of thinning.

Regarding instances of FP in the context of no glaucoma, when predominantly misclassified as glaucoma suspects, often exhibit a focus on vascular patterns resembling notching and the cup rim within the temporal region. These patterns may also be erroneously interpreted as indicative of glaucoma by the model, resulting in FP predictions for these cases.

Nonetheless, it is imperative to acknowledge the limitations inherent in these models, particularly with respect to the training phase. The inherent ambiguity and inconsistency in labeling within the 2/3 dataset by ophthalmologists can present challenges during model training, ultimately impacting its performance. Subsequent research and refinement of the training methodology are indispensable in addressing these limitations and enhancing the overall efficacy of crop-out-based augmentation methods in the realm of glaucoma screening.

## 6 Future works

Attention maps serve as valuable tools for gaining insight into the regions of interest and decision-making processes within vision transformers. However, to fully utilize their interpretability, it is essential to align these attention maps with the ISNT rules. According to this framework, the RNFL quadrants are divided into four pie-shaped sections: inferior, superior, nasal, and temporal. Additionally, the RNFL clock hours are typically divided into slices, often ranging from four to twelve. As depicted in Fig 2, our configuration consists of fourteen slices, exceeding the conventional twelve slices. In our future experimental approach, we will adjust the number of slices to ensure alignment with the RNFL clock hours. This presents a potential avenue for future research, wherein techniques can be developed to enhance the visualization of attention maps in vision transformers. Such enhancements may involve the exploration of visualization methods that offer more explicit and intuitive depictions of the attention patterns within the model. By enhancing the interpretability of attention maps, we can facilitate a deeper comprehension of the features and patterns contributing to the classification decisions made by vision transformers. Additionally, we plan to conduct an ablation study in the future to gain a deeper understanding of the inner workings of the model. This approach involves systematically removing or modifying individual components within the transformer architecture such as layers, attention mechanisms, or even specific subsets of training data. Our objective is to isolate and identify the impact of each component on the model's overall performance and decision-making process.

Moreover, we intend to explore strategies for improving performance and reducing FN and FP in the classification process. An intriguing direction for further investigation is the potential synergy between the "donut method" and the "pie method" to enhance glaucoma detection and classification. The "donut method" has demonstrated promising outcomes in terms of sensitivity and specificity, while the "pie method" aligns with the ISNT rule and enhances interpretability. By integrating these two methodologies, we aim to leverage their individual strengths to potentially achieve higher accuracy and resilience in glaucoma screening. The "donut method", with its capability to capture localized features and patterns, can complement

the "pie method" in pinpointing specific regions of interest for glaucoma detection. Concurrently, the "pie method" can provide valuable insights into the model's focus on clinically relevant areas.

## Supporting information

**S1 File. Dataset upon request.** The minimal GlauCUTU dataset is available upon request. (ZIP)

## Acknowledgments

Approval of all ethical and experimental procedures and protocols was granted by the Chulalongkorn Institutional Review Board under Application No. 715/61.

## Author contributions

**Conceptualization:** Sirikorn Sangchocanonta, Phongphan Phienphanich, Rath Itthipanichpong, Kitiya Ratanawongphaibul, Sunee Chansangpetch, Anita Manassakorn, Visanee Tantisevi, Prin Rojanapongpun, Charturong Tantibundhit.

**Data curation:** Phongphan Phienphanich, Rath Itthipanichpong, Kitiya Ratanawongphaibul, Sunee Chansangpetch, Anita Manassakorn, Visanee Tantisevi, Prin Rojanapongpun.

**Formal analysis:** Sirikorn Sangchocanonta, Charturong Tantibundhit.

**Funding acquisition:** Rath Itthipanichpong, Kitiya Ratanawongphaibul, Sunee Chansangpetch, Anita Manassakorn, Visanee Tantisevi, Prin Rojanapongpun, Charturong Tantibundhit.

**Investigation:** Sirikorn Sangchocanonta, Charturong Tantibundhit.

**Methodology:** Sirikorn Sangchocanonta, Charturong Tantibundhit.

**Project administration:** Charturong Tantibundhit.

**Resources:** Sirikorn Sangchocanonta, Adirek Munthuli, Sujittra Puangarom, Rath Itthipanichpong, Kitiya Ratanawongphaibul, Sunee Chansangpetch, Anita Manassakorn, Visanee Tantisevi, Prin Rojanapongpun, Charturong Tantibundhit.

**Software:** Sirikorn Sangchocanonta, Phongphan Phienphanich.

**Supervision:** Charturong Tantibundhit.

**Validation:** Sirikorn Sangchocanonta, Adirek Munthuli, Sujittra Puangarom, Rath Itthipanichpong, Kitiya Ratanawongphaibul, Sunee Chansangpetch, Anita Manassakorn, Visanee Tantisevi, Prin Rojanapongpun, Charturong Tantibundhit.

**Visualization:** Sirikorn Sangchocanonta.

**Writing – original draft:** Sirikorn Sangchocanonta.

**Writing – review & editing:** Pakinee Pooprasert, Nichapa Lerthirunvibul, Kanyarak Patchimnan, Adirek Munthuli, Rath Itthipanichpong, Charturong Tantibundhit.

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
