## [Decision Letter · Decision Letter 0]

27 Feb 2024

PONE-D-23-38529Optimizing Deep Learning Models for Glaucoma Screening with Vision Transformers for Resource Efficiency and the Pie Augmentation MethodPLOS ONE

Dear Dr. Itthipanichpong,

Thank you for submitting your manuscript to PLOS ONE. After careful consideration, we feel that it has merit but does not fully meet PLOS ONE’s publication criteria as it currently stands. Therefore, we invite you to submit a revised version of the manuscript that addresses the points raised during the review process.

We look forward to receiving your revised manuscript.

Kind regards,

Tariq Mahmood Khan, PhD

Academic Editor

PLOS ONE

4. Thank you for stating the following in the Acknowledgments Section of your manuscript: "This work was supported by the Health Systems Research Institute (HSRI)"

Please remove any funding-related text from the manuscript and let us know how you would like to update your Funding Statement. Currently, your Funding Statement reads as follows:"This work was supported by the Health Systems Research Institute (HSRI). The funder has no role in our study design, data collection and analysis, decision to publish, or preparation of the manuscript."

5. Thank you for stating the following in your Competing Interests section: "No"

6. We note that you have indicated that there are restrictions to data sharing for this study. For studies involving human research participant data or other sensitive data, we encourage authors to share de-identified or anonymized data. However, when data cannot be publicly shared for ethical reasons, we allow authors to make their data sets available upon request. For information on unacceptable data access restrictions, please see http://journals.plos.org/plosone/s/data-availability#loc-unacceptable-data-access-restrictions. 

Reviewers' comments:

Reviewer's Responses to Questions

**Comments to the Author**

1. Is the manuscript technically sound, and do the data support the conclusions?

Reviewer #1: Yes

Reviewer #2: Yes

2. Has the statistical analysis been performed appropriately and rigorously? 

Reviewer #1: Yes

Reviewer #2: Yes

3. Have the authors made all data underlying the findings in their manuscript fully available?

Reviewer #1: Yes

Reviewer #2: Yes

4. Is the manuscript presented in an intelligible fashion and written in standard English?

Reviewer #1: Yes

Reviewer #2: Yes

5. Review Comments to the Author

Reviewer #1: The work seems satisfactory, practical significance, innovation, novelty, scientific contribution , presented novel and better results and the work has the potential for acceptance and publication. Sufficient experiments have also been performed to validate the approach. It can be accepted after the incorporation given below certain suggestions properly.

1. Please discuss the time complexity of these DL models .

2. Please discuss the actual time taken by these DL models (in seconds).

3. Compare the results with prior state-of-the-art studies.

4. These recent state-of-the-art studies on glaucoma prediction should be discussed in the background, (Deep learning system applicability for rapid glaucoma prediction from fundus images across various data sets;Collaboration of features optimization techniques for the effective diagnosis of glaucoma in retinal fundus images;Efficient feature selection based novel clinical decision support system for glaucoma prediction from retinal fundus images;Emperor penguin optimization algorithm-and bacterial foraging optimization algorithm-based novel feature selection approach for glaucoma classification from fundus images)

5 . Please perform ablation study

6. Please share the dataset on any public repository and share the link so that other researchers can also validate their latest .

Regards,

Reviewer #2: The manuscript titled 'Optimizing Deep Learning Models for Glaucoma Screening with Vision Transformers for Resource Efficiency and the Pie Augmentation Method' is well-written overall. However, I have identified a few concerns that warrant attention.

1. The introduction sets a good foundation for the paper's objectives, but it could benefit from expanding on related work. It would be helpful to include references to studies like "Screening of Glaucoma disease from retinal vessel images using semantic segmentation," "A Review on Glaucoma Disease Detection Using Computerized Techniques," and "CDED-Net: Joint Segmentation of Optic Disc and Optic Cup for Glaucoma Screening" to give readers a broader context and show how this research fits into the existing body of work.

2. While the paper focuses on developing lightweight networks, it overlooks some important aspects. Specifically, there's a lack of detailed analysis on trainable parameters compared to existing methods. It would be valuable to include this analysis to demonstrate the efficiency gains achieved by the proposed approach. Additionally, examining training and testing times would give readers a better understanding of the practical implications of the research. By addressing these points, the paper could provide a more comprehensive evaluation of its contributions.

6. PLOS authors have the option to publish the peer review history of their article (what does this mean?). If published, this will include your full peer review and any attached files.

Reviewer #1: **Yes: **Munish Khanna

Reviewer #2: No

---

## [Author Response · Author response to Decision Letter 1]

21 Apr 2024

March 19, 2024

PLOS ONE

Dear Reviewers,

Thank you for taking the time to review our manuscript and for providing constructive feedback. We appreciate your positive remarks regarding the practical significance, innovation, novelty, scientific contribution, and potential for acceptance of our work. We have carefully reviewed your comments and suggestions and have addressed each one as outlined below:

Reviewer #1

1. Please discuss the time complexity of these DL models:

 In Section 2.3, we have included a detailed discussion on the time complexity of the deep learning (DL) models used in our study. We evaluated the time complexity of the ConvNeXt-S and DeiT-S models. Critical factors such as the computational resources required for training and their efficiency are reflected in this evaluation. As shown in Fig. 1, we illustrated the efficiency of each model in terms of ImageNet-1K accuracy and Giga floating point operations per second (GFLOPs) using the calflops library. ConvNeXt-S achieves a higher accuracy of 83.1%, potentially due to its complex architecture, comprising 50M parameters. However, such complexity demands more computational resources, resulting in higher time complexity and requiring 8.7 GFLOPs for inference. In contrast, DeiT-S prioritizes efficiency with its simpler architecture, consisting of 22M parameters. This results in a lower time complexity of 4.6 GFLOPs and a slightly lower accuracy of 79.8%. This compromise suggests that DeiT-S, while exhibiting slightly lower accuracy, offers a more efficient utilization of computational resources. Therefore, DeiT-S presents a particular advantage in resource-constrained environments where computational efficiency is of paramount importance.

2. Please discuss the actual time taken by these DL models (in seconds):

 We evaluated the real-world performance of the ConvNeXt-S and DeiT-S models. By measuring the actual time taken to complete the practical test of classifying a single fundus image and producing a heat map, we gain insights into their efficiency and suitability for deployment. Table 8 displays the results of the performance test conducted on both central processing unit (CPU) and graphics processing unit (GPU). The test on GPU portrays that ConvNeXt-S takes approximately 2 seconds, while DeiT-S takes merely 0.2 seconds, offering a tenfold reduction in the actual time required for classifying results and generating heat maps. On the CPU, both ConvNeXt-S’ and DeiT-S’ ability to complete tasks in less than a second, with DeiT-S being faster. The test on CPU is especially applicable for virtual machines in hospitals across various regions where resource consideration is critical. Hence, our proposed approach offers a significant reduction in computational time, enhancing the models' capability to classify results for individual fundus images expeditiously.

3. Compare the results with prior state-of-the-art studies:

 As shown in Table 7, our evaluation ensured that the proposed method was not trained on the DRISHTI-GS1 test set. This approach allows us to determine how well the model performs on unseen data. This allows us to compare our work's performance to previous studies and assess its generalizability. Our results are promising, achieving an F1-score of 93.33%, accuracy of 90.20%, and sensitivity of 93.11%. While there is a chance at improvement, these results indicate the generalizability and potential of our method for real-world applications in glaucoma classification.

4. Discussion of recent state-of-the-art studies, including references to related work:

 Following the reviewer's suggestion, we have added the introduction section with a discussion on recent studies related to glaucoma prediction. We have specifically addressed the four studies you mentioned to highlight the effectiveness of outcomes from optimized classification.

5. Ablation Study:

 Thank you for your suggestion. We are going to conduct an ablation study in the future to gain a deeper understanding of the inner workings of the model. This approach involves systematically removing or modifying individual components within the transformer architecture such as layers, attention mechanisms, or even specific subsets of training data. Our objective is to isolate and identify the impact of each component on the model’s overall performance and decision-making process.

6. Dataset Sharing:

 We recognize the value of data sharing for scientific progress and allowing the research community to verify findings. Unfortunately, due to ethical agreements with the hospital providing the data, we cannot publicly share it. This agreement restricts data use to our team to safeguard patient privacy and uphold ethical standards. Despite this, we have made every effort to describe our dataset and methodologies in detail within our manuscript to ensure as much transparency as possible. In consideration of this, we are committed to providing a minimal dataset upon request, which includes study IDs and predictions from our proposed model. Researchers interested in accessing this dataset can contact the IRB human research office at King Chulalongkorn Memorial Hospital, Thai Red Cross Society, using the following information:

 - Contact: IRB Office, 3rd Floor, Anandamahidol Building

 - Telephone: 0 2256 4493 / 098 573 7622

 - Email: medchulairb@chula.ac.th

 - Website: https://irb.md.chula.ac.th

Furthermore, we have conducted additional testing using the DRISHTI-GS1 public dataset, which our models have never been trained on. Testing on diverse datasets strengthens our findings' generalizability and validates our research.

Reviewer #2

1. Inclusion of references to related work:

 We have carefully reviewed the literature and incorporated references to the studies you suggested. These references are now included in a paragraph within the introduction section, which mentions that AI can automate various tasks such as cup and disc segmentation, vertical cup-to-disc ratio calculation in the optic nerve head (ONH), and relevant feature extraction.

2. The need for a more comprehensive evaluation of lightweight networks:

 In Section 2.3, we analyze the trainable parameters of our proposed model compared to existing methods to demonstrate its efficiency. Additionally, we discuss both the theoretical training times in Section 2.3 and the actual time taken for testing in the discussion and conclusion sections, providing a comprehensive picture of the model's computational requirements.

Finally, we really appreciated your valuable feedback and considering our work for publication.

Sincerely,

Rath Itthipanichpong (corresponding author)

E-mail: doctorrath@gmail.com

---

## [Decision Letter · Decision Letter 1]

6 Nov 2024

Optimizing Deep Learning Models for Glaucoma Screening with Vision Transformers for Resource Efficiency and the Pie Augmentation Method

PONE-D-23-38529R1

Dear Dr. Itthipanichpong,

We’re pleased to inform you that your manuscript has been judged scientifically suitable for publication and will be formally accepted for publication once it meets all outstanding technical requirements.

Kind regards,

Muhammad Mateen

Academic Editor

PLOS ONE

Additional Editor Comments (optional):

The authors addressed all the comments of reviewers carefully. Now, It is recommended for publication.

Reviewers' comments:

Reviewer's Responses to Questions

**Comments to the Author**

1. If the authors have adequately addressed your comments raised in a previous round of review and you feel that this manuscript is now acceptable for publication, you may indicate that here to bypass the “Comments to the Author” section, enter your conflict of interest statement in the “Confidential to Editor” section, and submit your "Accept" recommendation.

Reviewer #1: All comments have been addressed

Reviewer #3: All comments have been addressed

2. Is the manuscript technically sound, and do the data support the conclusions?

Reviewer #1: Yes

Reviewer #3: Yes

3. Has the statistical analysis been performed appropriately and rigorously? 

Reviewer #1: Yes

Reviewer #3: Yes

4. Have the authors made all data underlying the findings in their manuscript fully available?

Reviewer #1: Yes

Reviewer #3: No

5. Is the manuscript presented in an intelligible fashion and written in standard English?

Reviewer #1: Yes

Reviewer #3: Yes

6. Review Comments to the Author

Reviewer #1: This revised version can be updated now.

Congratulations to all authors !!

Reviewer #3: Previous reviewers observations have been addressed and these have been highlighted by the authors. I believe the article can now accepted for publication,

7. PLOS authors have the option to publish the peer review history of their article (what does this mean?). If published, this will include your full peer review and any attached files.

Reviewer #1: **Yes: **Munish Khanna

Reviewer #3: No

---

## [Editor Report · Acceptance letter]

PONE-D-23-38529R1

PLOS ONE

Dear Dr. Itthipanichpong,

I'm pleased to inform you that your manuscript has been deemed suitable for publication in PLOS ONE. Congratulations! Your manuscript is now being handed over to our production team.

Kind regards,

on behalf of

Dr. Muhammad Mateen

Academic Editor

PLOS ONE